behaviour, neuroscience, ecology

distributed visual systems, neuroethology, visual ecology, sensory processing, circular statistics

**Author for correspondence:**
Daniel R. Chappell
e-mail: daniel.r.chappell@gmail.com

# Panoramic spatial vision in the bay scallop *Argopecten irradians*

Daniel R. Chappell, Tyler M. Horan and Daniel I. Speiser

Department of Biological Sciences, University of South Carolina, 715 Sumter Street, Columbia, SC 29208, USA

DRC, 0000-0003-2214-9335; DIS, 0000-0001-6662-3583

We have a growing understanding of the light-sensing organs and light-influenced behaviours of animals with distributed visual systems, but we have yet to learn how these animals convert visual input into behavioural output. It has been suggested they consolidate visual information early in their sensory-motor pathways, resulting in them being able to detect visual cues (spatial resolution) without being able to locate them (spatial vision). To explore how an animal with dozens of eyes processes visual information, we analysed the responses of the bay scallop *Argopecten irradians* to both static and rotating visual stimuli. We found *A. irradians* distinguish between static visual stimuli in different locations by directing their sensory tentacles towards them and were more likely to point their extended tentacles towards larger visual stimuli. We also found that scallops track rotating stimuli with individual tentacles and with rotating waves of tentacle extension. Our results show, to our knowledge for the first time that scallops have both spatial resolution and spatial vision, indicating their sensory-motor circuits include neural representations of their visual surroundings. Exploring a wide range of animals with distributed visual systems will help us learn the different ways non-cephalized animals convert sensory input into behavioural output.

## 1. Introduction

A diverse set of invertebrates have many separate light-sensing organs distributed across their bodies. The light-sensing organs that contribute to these distributed visual systems range from pigment-shielded photoreceptors like those of the brittle star *Ophiocoma* [1], to eyespots like those of chitons such as *Chiton* [2], to compound eyes like those of sabellid and serpulid fan worms [3], to camera-type eyes like those of cubozoans such as *Tripedalia* [4] and chitons such as *Acanthopleura* [5,6]. Many animals with distributed visual systems display behaviours that require spatial information about light. These animals include, but are not limited to, sea urchins such as *Strongylocentrotus* [7] and *Diadema* [8], brittle stars such as *Ophiocoma* [9], sea stars such as *Linckia* [10] and cubozoans such as *Tripedalia* [11,12]. From these recent studies, we have a growing understanding of the light-sensing organs and light-influenced behaviours of animals with distributed visual systems, but we have yet to learn how the sensory-motor circuits of these animals convert sensory input into behavioural output.

Studying sensory-motor circuits in animals with distributed visual systems has been challenging because the nervous systems of these animals tend to be less centralized than those of animals with cephalic eyes [13–15]. Extracting spatial information from visual input is thought to require a brain, so it has been hypothesized that animals with distributed visual systems consolidate visual information early in their sensory-motor pathways [16]. The degree to which animals with distributed visual systems consolidate visual input may vary, with animals that consolidate to a greater degree having less ability to extract spatial information. One possibility is that animals consolidate visual

information at the level of their entire visual system. In this scenario, an animal will detect visual cues without being able to locate them, i.e. the animal will have spatial resolution, but not spatial vision [16,17]. A second possibility is that animals consolidate visual information at the level of their light-sensing organs and extract spatial information from comparisons between light-sensing organs (interocular spatial vision). In this scenario, animals will be able to locate visual cues in relation to their bodies, but not in relation to their eyes. A third possibility is that animals do not consolidate spatial information at the level of their light-sensing organs and extract spatial information from comparisons between photoreceptors from the same light-sensing organs (intraocular spatial vision). In this scenario, animals will be able to locate visual cues in relation to their bodies and their eyes.

Scallops, a type of bivalve mollusc (family Pectinidae), are a promising model for studying how animals with distributed visual systems consolidate visual information in their sensory-motor circuits. Scallops have a distributed visual system that includes dozens of eyes arrayed along the edges of their valves. These eyes have fields of view of 90–100° and each eye forms an image, primarily through the reflection of light by a concave mirror [18]. Physiological studies indicate the eyes of scallops respond to rotating stripes with angular widths as narrow as 2° [19], a finding consistent with estimates of visual acuity from computational models [20,21] and behavioural tests [22,23]. Despite strong evidence that scallops have spatial resolution, we lack evidence that these animals have spatial vision. It remains possible, therefore, that scallops consolidate visual information so that they are able to detect stimuli, such as moving objects, without being able to locate them in their visual field.

To ask how scallops consolidate visual information obtained by their many image-forming eyes, we compared how the bay scallop *Argopecten irradians* extends its sensory tentacles towards static visual stimuli with different locations and sizes, as well as rotating isoluminant visual stimuli of different sizes. The sensory tentacles of scallops are highly flexible, and they are interspersed with the eyes along the edges of both of the valves (figure 1). Scallops use their sensory tentacles to identify potential predators based on tactile and chemical information [24,25]. Scallops have been observed extending their sensory tentacles towards visual cues [22], but these behaviours have yet to be quantified. Tentacle extension behaviours are useful for exploring the sensory-motor circuits of scallops because they provide an external reference of where a scallop perceives an object to be in relation to its own body. By studying the visually guided tentacle extension behaviours of scallops, we will gain insight into how these many-eyed animals construct neural representations of their environment.

## 2. Material and methods

### (a) Specimen acquisition and care
We acquired *A. irradians* from Gulf Specimen Marine Laboratory (Panacea, FL). These specimens had shell heights ranging from 3.5 to 5.7 cm. We kept *A. irradians* at the University of South Carolina (Columbia, SC) in a Living Stream System (Frigid Units, Toledo, OH) with recirculating natural seawater (NSW)

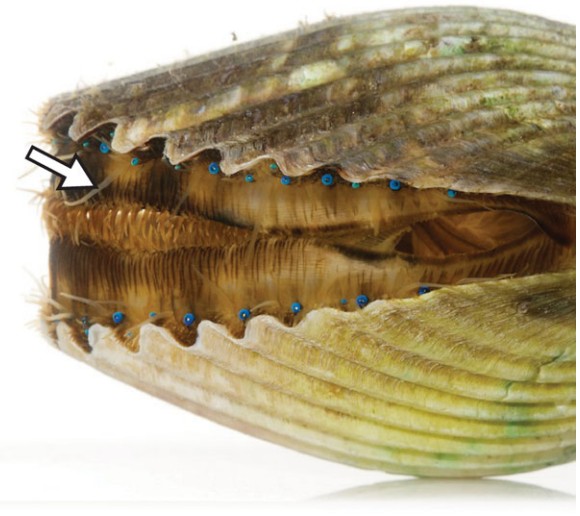

**Figure 1.** The bay scallop *Argopecten irradians* from a posterior view with the ventral edges of its valves facing left. The mantle tissues are adorned with numerous striking blue eyes and a variety of tentacles. The large sensory tentacles that scallops can extend towards visual stimuli are the primary sensory tentacles (one is labelled by an arrow). The other tentacles are shorter and less extensible. These include the secondary sensory tentacles, which are interspersed with the primary sensory tentacles, and the guard tentacles, which are seen interlocked on the muscular curtain that closes off the gape between the valves. Photo credit: David Liittschwager. (Online version in colour.)

held at a temperature of 19.5°C and a salinity of 33 ppt. For lighting, we used two Hydra TwentySix HD LED fixtures (AquaIllumination, Ames, IA) set to a light/dark cycle of 12 : 12 h. We fed scallops three times per week with 120 ml of DT's Live Marine Phytoplankton Premium Reef Blend (Sustainable Aquatics, Jefferson City, TN).

### (b) Equipment for behavioural trials
For trials with static visual stimuli, we used a clear acrylic cylinder (20 × 20 cm) filled with NSW as a behavioural arena. We placed a second clear acrylic cylinder (25 × 25 cm) so that the smaller cylinder was enclosed by the larger one. We wrapped the outside of the larger cylinder with white paper on which we had printed visual stimuli. We positioned the two cylinders on top of a clear acrylic stand and recorded scallops from below using an HD C615 webcam (Logitech International, Newark, CA). We lit the behavioural arena from above using an Aqua Illumination Prime HD LED fixture (C2 Development, Inc., Ames, IA). We diffused light from this fixture with two filters mounted in series (3000 Tough Rolux and 3027 Half Tough White Diffusion; Rosco Laboratories, Stamford, CT, USA).

For trials with rotating visual stimuli, we used a clear acrylic cylinder (20 × 20 cm) filled with NSW as a behavioural arena. Around this first cylinder, which remained motionless, we rotated a second clear acrylic cylinder (30 × 30 cm) wrapped with white paper on which we had printed visual stimuli. To rotate the outer cylinder, we used a stepper motor (OMC Corporation Limited, Nanjing City, China) operated by an Arduino Uno microcontroller (Arduino LLC, Somerville, MA, USA) with an attached motor shield (Adafruit, New York, NY, USA). We recorded scallops from above using a GoPro Hero 6 (GoPro Inc., San Mateo, CA, USA) at a frame rate of 30 fps. We lit the behavioural arena from above and below using two Aqua Illumination Prime HD LED fixtures (C2 Development, Inc., Ames, IA). We diffused light from these using two filters mounted in series (3000 Tough Rolux and 3027 Half Tough White Diffusion; Rosco Laboratories, Stamford, CT, USA).

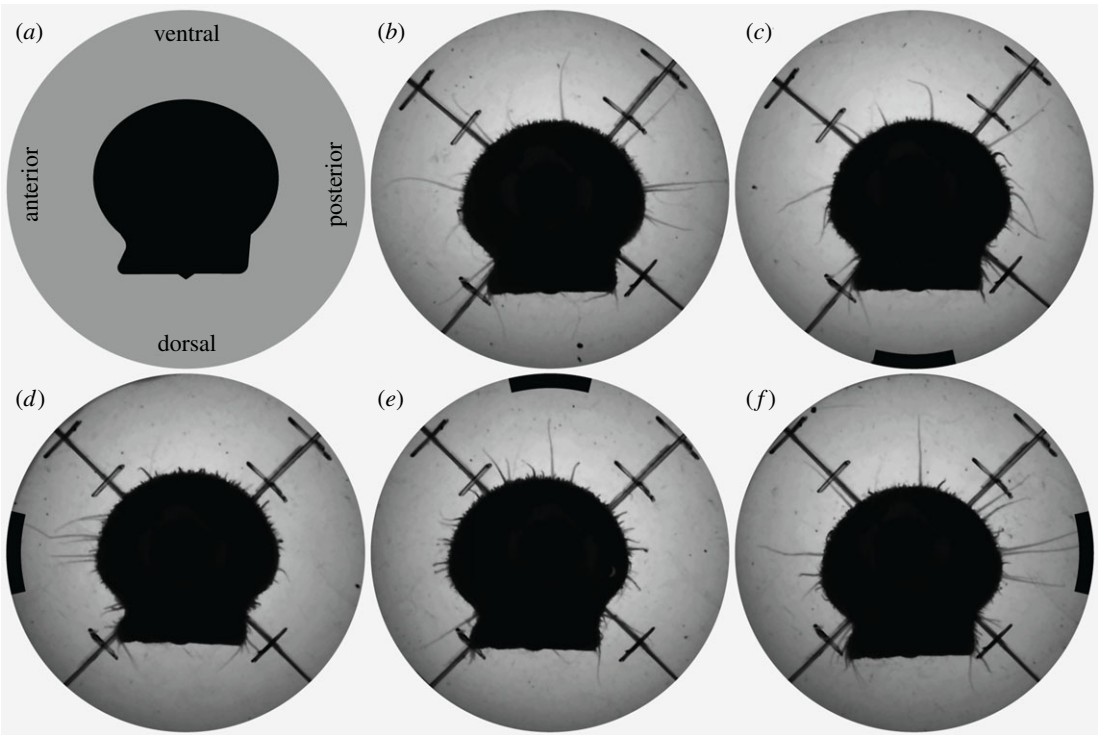

**Figure 2.** A bay scallop extending its sensory tentacles in response to black stripes presented at different angular locations. The sensory tentacles appear as light grey lines extending from the valve margins. (*a*) The body axes of the scallop. (*b–f*) Still frames from videos showing the scallop's responses to (*b*) a control stimulus and dark stripes with angular widths of 24° that we presented (*c*) dorsally, (*d*) anteriorly, (*e*) ventrally and (*f*) posteriorly. In (*c–f*), the curved black bars represent the positions and relative angular widths of the visual stimuli we presented.

## (c) Procedures for behavioural trials

To begin trials with static visual stimuli, we placed a scallop in the middle of the behavioural arena and then positioned the outer cylinder to which we had affixed the visual stimulus for the trial. We emptied and replaced the NSW in the behavioural arena every three trials. In our first experiment with static visual stimuli, we recorded the responses of scallops (*n* = 21) to five treatments. These included a control treatment consisting of an isoluminant white stimulus and four experimental treatments in which we presented a single black stripe with an angular width of 24° (as measured from the centre of the behavioural arena). We displayed this stripe at four different positions relative to test animals (figure 2): ventral (0°), posterior (90°), dorsal (180°) or anterior (270°). We presented all five treatments in a random order to every animal so that each animal experienced one treatment per day.

In our second experiment with static visual stimuli, we recorded the responses of scallops (*n* = 21) to four treatments. These included a control treatment consisting of an isoluminant white stimulus and three experimental treatments in which we presented a single black stripe to the ventral sides (0°) of test animals. These stripes had angular widths of either 6, 12 or 24° (as measured from the centre of the behavioural arena). We presented all four treatments in a random order to every animal so that each animal experienced one treatment per day.

To begin trials involving rotating visual stimuli, we placed a scallop in the middle of the behavioural arena and then positioned the outer cylinder to which we had affixed the visual stimulus for the trial. We emptied and replaced the NSW in the behavioural arena every three trials. We recorded the responses of scallops (*n* = 16) to five treatments. These included a control treatment consisting of an all-grey stimulus and four isoluminant experimental treatments that included a black stripe flanked by white stripes against a grey background. The black and white stripes combined had angular widths of either 2, 5, 10 or 20° (as measured from the centre of the behavioural arena). The grey backgrounds had reflectance values half-way between the

reflectance values of the black and white stripes. We verified this using a portable fibre-optic spectrophotometer system from Ocean Optics (Dunedin, FL, USA) in which a Y-shaped reflection probe (QR400–7-UV-VIS) supplied light from a 20 W tungsten halogen lamp (HL-2000-HP-FHSA) and carried reflected light to a Flame-S-VIS-NIR-ES spectrometer operated using Ocean View software. We rotated the visual stimuli for one full rotation at a rate of approximately 0.5 r.p.m. We presented all five treatments in a random order to every animal so that each animal experienced one treatment per day. We randomized both the starting positions of the stripes relative to the body positions of test subjects and the direction at which the stripes rotated.

## (d) Data analysis for trials with static visual stimuli

To quantify the responses of *A. irradians* to static visual stimuli, we analysed a still frame from the digital recording of each trial taken from 30 s after a scallop began extending its tentacles. We chose this time point after observing the responses of *A. irradians* to visual cues in preliminary trials, but before analysing the results of any of the experimental trials reported here. We used FIJI [26] to acquire the coordinates of landmarks from each still frame. These landmarks included the centre of the test animal, the position of the visual stimulus and the tip and base of each extended sensory tentacle. From the coordinates we acquired, we calculated the relative lengths and angular directions of tentacles extended by test animals. We did not record the coordinates of tentacles that scallops did not extend during trials. We excluded from analysis trials in which scallops extended fewer than three tentacles.

We used R to implement four statistical analyses of the tentacle extension behaviours of *A. irradians*. First, to test if the populations of tentacles extended by scallops showed significant directedness, we used Moore's modified Rayleigh test [27]. Directedness is a measure of how closely the headings of a group of vectors are aligned. It is lowest when the headings of vectors are random and highest when the headings of vectors are in the same direction. Moore's modified Rayleigh test

yields metrics for the directedness of a group of vectors ($R^*$) and the mean angle of their directedness ($\phi^*$). We treated each tentacle extended by a scallop as a vector with a heading and a length, and all of the tentacles extended by a scallop during a trial as a group of vectors. We transformed the angles of extended tentacles relative to the positions of test animals so that 0, 90, 180 and 270° corresponded to the ventral, posterior, dorsal and anterior sides of scallops, respectively. Because scallops have dozens of tentacles and test animals extended different numbers of them during trials, the critical $R^*$ values for 95% likelihood differed between trials [27]. Also, we did not analyse trials in which scallops did not extend their tentacles, so sample sizes varied between treatments even though we tested the same number of scallops in each treatment. Several factors contributed to imprecision in our calculations of $\phi^*$ values. These included scallops extending their tentacles to different lengths during trials and the tentacles being distributed non-uniformly along the edges of the valves.

Second, we tested if the $\phi^*$ values of the populations of tentacles extended by scallops are better described by uniform or unimodal models of circular distribution. To do so, we used 'CircMLE' [28], an R package that uses a model selection procedure to rank how well models describe datasets. For each treatment in our experiments, we used $\phi^*$ values from trials as input into CircMLE to compare support between a null model of uniform circular distribution and an alternative model of unimodal circular distribution. The outputs of CircMLE included metrics of model performance (AICc, ΔAICc and AICc weights), the principle direction of the unimodal model ($\phi_1$) and the concentration parameter of the circular distribution of the unimodal model ($\kappa_1$). We used corrected Akaike information criterion (AICc) tests [29] to assess support for models owing to small sample sizes ($n < 20$) and we defined good support as ΔAICc scores of less than 2.00 [30,31]. To test for the homogeneity of the concentration parameters of the circular distributions ($\kappa_1$), we used the 'equal.kappa.test' function of the R package 'circular' [32]. For all statistical tests, we assessed significance with a critical value of $p < 0.05$.

Third, to test if the numbers of tentacles extended by scallops differed between the treatments within our two experiments with static visual stimuli, we used two (one for each experiment) omnibus Friedman rank sum tests of the form: (extended tentacle count) ~ (treatment) | (scallop identity). We assessed the significance of the tests using a critical value of $p < 0.05$.

Fourth, we tested if the tentacles extended by scallops exhibited higher values of directedness ($R^*$) when their mean angles of directedness ($\phi^*$) were towards the visual stimuli. To do so, we first transformed the angular positions of the visual stimuli and the mean angles of directedness of tentacles ($\phi^*$) from polar coordinates into Cartesian coordinates based on a unit circle. Then, we calculated the absolute difference in the x coordinate ($\Delta x$), the absolute difference in the y coordinate ($\Delta y$) and the linear distance between these points ($z$). We used these variables to construct a multiple linear regression model of the form: $R^* = 1 + \beta 1 + \beta 2$ ($z$). We fitted this model to the dataset using the 'lm' function in R which uses ordinary least-squares fitting, evaluated the model fit using an $F$-test with a critical value of $p < 0.05$, measured the variance explained by the model using the statistical metric $R^2$, assessed the normality of the residuals of the model using a Shapiro–Wilk test of normality with a critical value of $p > 0.05$ to indicate normally distributed residuals and calculated the confidence intervals of the model parameters using the 'confint' function in the 'stats' R package.

## (e) Data analysis for trials with rotating visual stimuli

To quantify the responses of *A. irradians* to rotating visual stimuli, we first downsampled the frame rate of the digital recording from 30 fps to 1 fps which resulted in each recording being approximately 130 frames long. We used FIJI [26] to extract coordinates of landmarks from each frame of the downsampled digital recordings. These landmarks included the centre of the test animal, the position of the visual stimulus and the tip and base of each extended sensory tentacle. We did not record the coordinates of non-extended tentacles. We excluded from analysis trials in which scallops did not extend tentacles. From the coordinates we acquired, we calculated the relative lengths and angular directions of tentacles extended by test animals.

We used R to analyse the tentacle extension behaviours of *A. irradians* in response to rotating visual stimuli. To quantify the directedness of the population of tentacles extended by a scallop during a trial, we first calculated the average length of every tentacle extended by a scallop over the 130 separate frames of each trial. We then used this metric to calculate the relative length of every extended tentacle in every frame. We paired the relative length of every extended tentacle with its angular direction to generate a population of vectors that described the population of tentacles extended by a scallop at every frame in a trial. We then fitted a von Mises unimodal distribution equation to this dataset, which yielded a mean direction ($\phi_1$) and concentration parameter ($\kappa_1$) for the population of tentacles extended by a scallop during a trial. We transformed the vectors of mean direction ($\phi_1$) and concentration parameter ($\kappa_1$) into Cartesian coordinates, which allowed us to quantify and plot how the circular distribution of relative tentacle lengths changed over the time course of trials.

## 3. Results

## (a) Bay scallops differentiate between static visual stimuli by location

In our first experiment, we found *A. irradians* differentiate between static visual stimuli presented in different locations by directing their sensory tentacles towards them (figure 2).

In our first analysis, we found tentacles extended by individual bay scallops showed significant values of directedness ($R^*$) in 4 out of 17 trials with control stimuli, 8 out of 20 trials with ventral stimuli, 5 out of 17 trials with anterior stimuli, 5 out of 18 trials with posterior stimuli and 0 out of 16 trials with dorsal stimuli.

Second, we found the mean angles of directedness ($\phi^*$) of the tentacles extended by bay scallops in response to visual stimuli in the anterior, ventral and posterior positions were described with strong support by unimodal models of circular distribution and with poor support by uniform models (electronic supplementary material, table S1). In these unimodal models, the principle directions ($\phi_1$) of the extended tentacles were towards the positions of the visual stimuli (figure 3; electronic supplementary material, table S1). Conversely, the $\phi^*$ values of the tentacles extended by scallops in response to the control stimulus and the dorsal visual stimulus were described with strong support by *both* unimodal and uniform models of circular distribution (figure 3; electronic supplementary material, table S1).

Third, we found bay scallops extended different numbers of tentacles during trials (0–32 tentacles per animal), but the numbers of tentacles they extended did not depend on the presence or location of visual stimuli ($p > 0.05$).

Fourth, we found the tentacles extended by bay scallops exhibited higher values of directedness ($R^*$) in trials in which the linear distance ($z$) between the visual stimulus

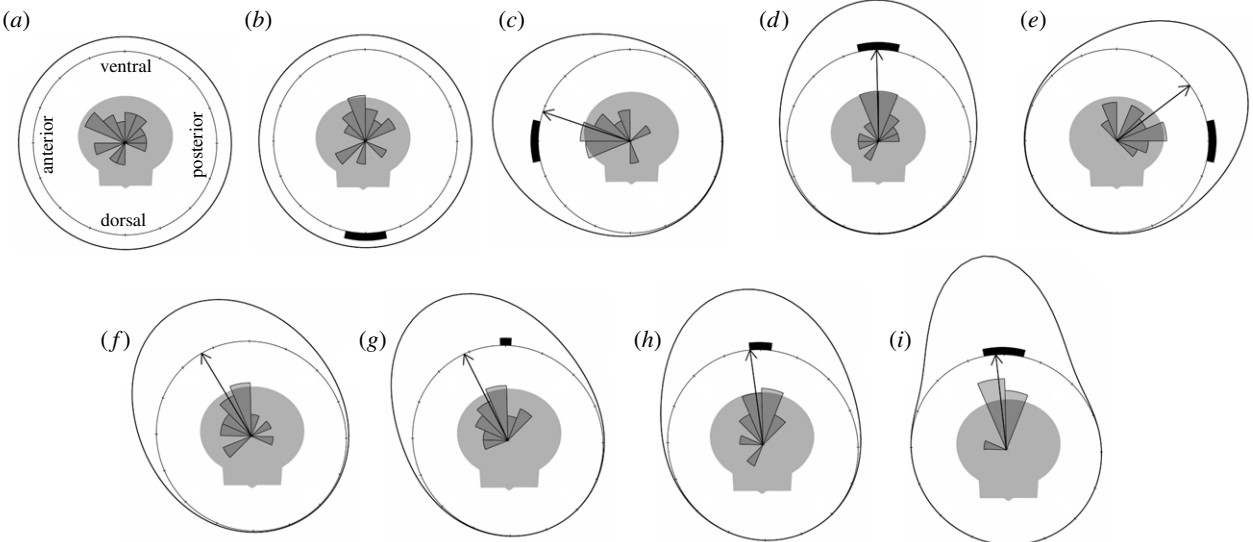

**Figure 3.** A summary of tentacle extension behaviours by bay scallops ($n = 21$) in response to static visual stimuli. ($a$–$i$) Circular histograms showing the distributions of the mean angles of tentacles extended by scallops in response to dark stripes that varied in location ($a$–$e$) and angular width ($f$–$i$). For each of the histograms, the radial space is divided into 16 bins so that each discrete bin corresponds to 22.5°. The plots include grey silhouettes indicating the positions of scallops during trials and curved black bars representing the positions and relative angular widths of the visual stimuli we presented. In ($a$–$i$), the outer black lines indicate the distributions of either the null models (i.e. uniform circular distribution) or the alternative models (i.e. unimodal circular distribution) and the black arrows indicate the principle directions ($\phi_1$) for the alternative models. In ($a$,$b$), the null model distributions are shown because their strong support did not allow us to accept the alternative model; in ($c$–$i$), the alternative model distributions are shown because weak support for the null model allowed us to reject it.

and the mean angle of the directedness of the extended tentacles ($\phi^*$) was smaller. After fitting our linear regression model to the results of our trials, we found the optimal coefficients of the model were $\beta_1 = 1.21$, $\beta_2 = -0.23$. We found that the residuals of the model were normally distributed (Shapiro–Wilks $p > 0.05$). The model significantly described our dataset ($F_{1,68} = 20.01$, $p < 0.05$), explained 23% of the variance we observed ($R^2 = 0.23$), and the $\beta_2$ parameter had a confidence interval between $-0.34$ and $-0.13$, which means that the value of directedness ($R^*$) and the linear distance ($z$) had a significantly inverse relationship. In other words, the tentacles extended by scallops exhibited higher values of directedness towards the visual stimuli than away from them.

## (b) Bay scallops differentiate between static visual stimuli by size

In our second experiment, we found *A. irradians* distinguish between static visual stimuli of different angular sizes by extending their tentacles in a more directed manner towards stimuli with greater angular widths.

In our first analysis, we found tentacles extended by individual bay scallops showed significant values of directedness ($R^*$) in 1 out of 19 trials with control stimuli, 8 out of 15 trials with stimuli 6° wide, 6 out of 17 trials with stimuli 12° wide and 9 out of 18 trials with stimuli 24° wide.

Second, we found the mean angles of directedness ($\phi^*$) of the tentacles extended by bay scallops in response to the control stimulus and to all three visual stimuli were described with strong support by unimodal models of circular distribution and with poor support by uniform models (electronic supplementary material, table S1). In these unimodal models, the principle directions ($\phi_1$) of the circular

distributions were ventral in all treatments, which corresponded to the position of the visual stimuli when they were present (figure 3; electronic supplementary material, table S1). Using an initial global test, we found a significant difference between the concentration parameters ($\kappa_1$) of these unimodal models ($p > 0.05$). Following with pairwise comparisons, we found the concentration parameters of the unimodal models of circular distribution for trials with the 24° stimulus were significantly greater than the concentration parameters of the unimodal models of circular distribution for trials with the control stimulus ($p < 0.05$), the 6° stimulus ($p < 0.05$) and the 12° stimulus ($p < 0.05$). From this, we conclude more scallops directed their extended tentacles towards the largest visual stimulus than towards either the negative control or the smaller visual stimuli.

Third, we found that bay scallops extended different numbers of tentacles during trials (ranging from 0 to 21 tentacles per animal), but the numbers of tentacles they extended did not depend on the presence or size of visual stimuli ($p > 0.05$).

## (c) Bay scallops track rotating visual stimuli with their tentacles

Scallops track rotating visual stimuli with rotating waves of tentacle extension. The rotation of the wave of tentacle extension displayed by a scallop was represented in our data as rotating mean angles ($\phi_1$) of the unimodal models fitted to values of relative tentacle extension (figure 4). In addition, the amplitude of the rotating wave of tentacle extension was represented as changes in the concentration parameters ($\kappa_1$) of these unimodal models (figure 4). When a scallop did not respond to a rotating visual stimulus, the mean

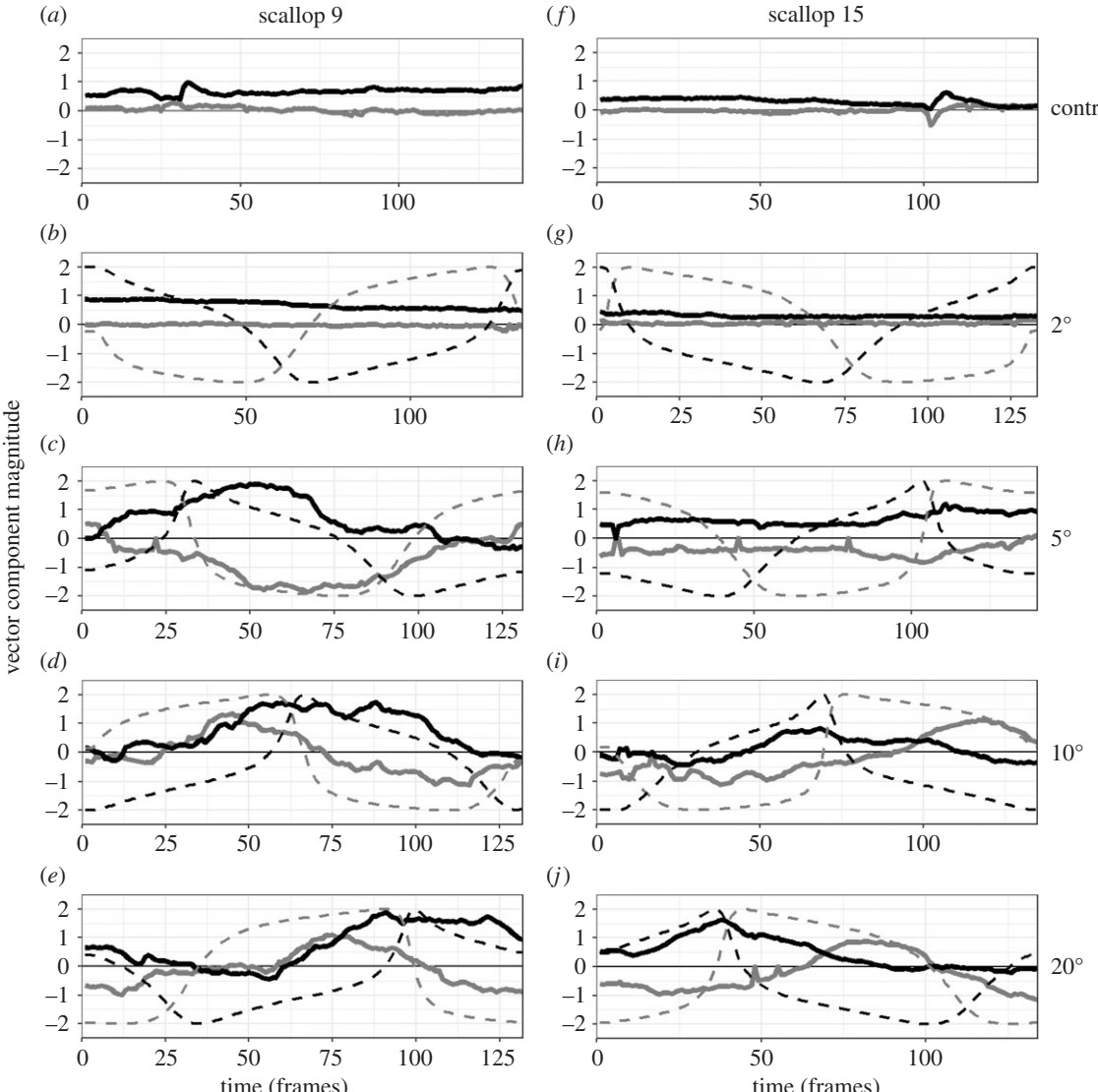

**Figure 4.** The tentacle extension responses of scallops to rotating isoluminant visual stimuli. The tentacle responses of scallop 9 (*a–e*) and scallop 15 (*f–j*) to the control stimulus (*a,f*), the 2° stimulus (*b,g*), the 5° stimulus (*c,h*), the 10° stimulus (*d,i*) and the 20° stimulus (*e,j*). Each line plot shows the *x*-coordinate (solid grey line) and the *y*-coordinate (solid black line) of the Cartesian transformed vector of the unimodal model of the circular distribution of relative tentacle length across each of the approximately 130 video frames in our trials. Each line plot also includes the *x*-coordinate (dashed grey line) and *y*-coordinate (dashed black line) of the Cartesian transformed angular location of the rotating visual stimulus. We consider scallops to have followed a stimulus if their unimodal distribution of extended tentacles rotated with the stimulus. Accordingly, scallop 9 tracked the 5, 10 and 20° rotating stimuli (*c,d,e*) and scallop 15 tracked the 10 and 20° stimuli (*i,j*). All of the behavioural trials corresponding to (*a–j*) have been included as electronic supplementary material, videos S1–S10.

angles ($\phi_1$) and concentration parameters ($\kappa_1$) of the unimodal models were relatively constant (e.g. figure 4*a*). By contrast, when a scallop tracked a rotating visual stimulus with its extended tentacles, the mean angles of the unimodal models ($\phi_1$) coarsely tracked the angular position of the visual stimulus (e.g. figure 4*e*). Out of the 16 scallops that viewed each rotating visual stimulus, we observed this tracking behaviour in 11 animals viewing the 20° stimulus, eight viewing the 10° stimulus, nine viewing the 5° stimulus, none viewing the 2° stimulus and none viewing the negative control. When a scallop tracked a rotating visual stimulus, we also saw large concentration parameters ($\kappa_1$) which indicated greater unimodal distributions of relative tentacle length (electronic supplementary material, videos S1–S10). In many trials, the mean angles of the unimodal models ($\phi_1$) lagged behind the angular position of the visual stimulus, which can be explained by the tentacles extending and retracting relatively slowly.

## 4. Discussion

### (a) Bay scallops demonstrate spatial resolution and spatial vision

We found that bay scallops differentiate between visual stimuli based on their locations and sizes and track moving visual stimuli with their sensory tentacles. In our first experiment, bay scallops responded to visual stimuli presented in anterior, ventral and posterior positions by directing their extended sensory tentacles towards them. These results are consistent with previous observations that scallops extend their sensory tentacles towards visual stimuli [22,24,25]. They also indicate that bay scallops have panoramic spatial vision: without moving their bodies, these animals are able to locate visual cues spanning an arc around their valves of at least 270°. In our second experiment, bay scallops differentiated between visual stimuli that varied in angular width.

They did so by more frequently directing their extended sensory tentacles towards wider visual stimuli. In our third experiment, bay scallops indicated they can track moving objects by responding to rotating isoluminant visual stimuli with rotating waves of tentacle extension. Together, our results confirm prior reports that scallops have spatial resolution [19,20–23] and, for the first time, to our knowledge, demonstrate that scallops have spatial vision.

Despite their lack of cephalization, bay scallops are able to both detect *and* locate visual cues using their distributed visual system. How do they accomplish this task? It has been proposed that animals with distributed visual systems simplify the processing of visual information by consolidating spatial input from their light-sensing organs early in their sensory-motor circuits [16]. Contrary to this prediction, our experiments suggest bay scallops do not consolidate all of the spatial information they obtain with their image-forming eyes. Instead, they retain spatial information, create a neural representation of their visual field and then extract spatial information from this internal map to direct their sensory tentacles towards visual stimuli.

The results of our experiments are consistent with a model of visual processing in which bay scallops extract spatial information from their distributed visual system by comparing visual input between their eyes (interocular spatial vision). Our results also suggest that bay scallops may extract spatial information from their surroundings by comparing visual input between photoreceptors from the same eye (intraocular spatial vision). In the absence of a visual stimulus, bay scallops extend their sensory tentacles so that they point outwards from the valve margins in a perpendicular orientation (figure 2*b*). However, in the presence of a visual stimulus, bay scallops point their extended sensory tentacles at the stimulus so that the tentacles are no longer perpendicular to the valve margins (electronic supplementary material, video S11). By doing so, scallops indicate they can assess the locations of visual stimuli relative to the positions of their sensory tentacles. To do so, scallops require spatial vision finer than that probably made possible by interocular comparisons. Scallops can only acquire coarse spatial information from interocular comparisons because their eyes have wide (90–100°) and highly overlapping fields of view [18]. Consequently, we propose that scallops use intraocular comparisons to extract spatial information from their surroundings, which may provide finely grained spatial vision because their eyes form images with angular resolutions as fine as 2° [20,21].

## (b) Sensory ecology of scallops: cross-modal predator detection and identification

Unlike most bivalves, many species of scallop are able to swim using a form of jet propulsion [33,34]. Swimming may help scallops evade certain types of predators, but it is a risky behaviour. After a single bout of swimming, scallops must recover and during this time they are vulnerable to attack [35]. Consequently, it will benefit scallops to reliably distinguish between non-threats and threats because swimming in response to encounters with non-predatory animals may prevent scallops from escaping predators they encounter shortly thereafter. Scallops assess potential predators by gathering tactile and chemical information with their sensory tentacles [36]. Using their sensory tentacles, bay scallops can distinguish between predatory and non-predatory snails and between predatory sea stars and non-predatory sea urchins [36]. Scallops even vary the magnitudes of their swim responses based on the identities of predators [36,37].

We hypothesize that scallops use their eyes and sensory tentacles together as part of a two-step system for detecting and identifying predators. We propose that scallops use spatial vision for long-range predator detection and then use their sensory tentacles for short-range predator identification. For scallops to identify predators with their sensory tentacles, they must obtain sufficient tactile and chemical information. By using spatial vision to direct their sensory tentacles towards objects, bay scallops will be able to sample from larger areas of potential predators for longer periods of time. By gathering more chemo-tactile information, scallops will increase the likelihood they successfully identify a predator and initiate an escape response. If bay scallops detect potential predators using spatial vision and then identify predators using chemical and tactile cues, it would indicate that even animals with distributed visual systems can employ cross-modal links to guide their overt spatial attention [38].

## (c) Central and peripheral processing of visual information in the scallop nervous system

Bay scallops use spatial input from their distributed visual system to locate and distinguish the sizes of static objects and to follow the positions of moving objects. It is surprising to see such complex visual-motor behaviours in a non-cephalized animal with a distributed visual system, in part because the visual-motor circuits that coordinate such behaviours tend to be located in the brains of animals with cephalized sensory systems [39]. We propose the components of the visual-motor circuits of scallops are found in both their ganglion-based central nervous system and in the medullary cords of their peripheral nervous system.

Past findings indicate that at least some of the components of the visual-motor circuits of scallops are located in their visceral ganglion (VG), a nerve centre on the ventral surface of their adductor muscle. The optic nerves exiting the eyes of scallops project to the cortical surfaces of the paired lateral lobes of the VG where they make synaptic contact with first-order interneurons [40]. These interneurons project neurites that contribute to glomerular neuropil structures embedded within the lateral lobes [40]. Exposing the eyes of scallops to light elicits electrical activity on the cortical surfaces of the lateral lobes, but it remains unknown how light influences activity in the glomerular structures [41].

In addition to the lateral lobes of the VG, it is possible that the peripheral nervous system also contributes to visual-motor processing. Scallops have two circumpallial nerves which separately circle the mantle margins of their left (upper) and right (lower) valves. Intriguingly, the circumpallial nerves are not strictly nerves, but rather medullary cords with an outer layer of cell bodies and an inner layer of neuropil [42]. Thus, there may be neural circuits within the circumpallial nerves of scallops. Previous experiments have discounted the possibility of visual-motor processing in the circumpallial nerves because the optic nerves appear to remain isolated from the other neurons that are present [40]. However, even if the circumpallial nerves do not contribute to visual processing, they may still play an important role in motor control of the mantle tissues including the sensory

tentacles. Our hypothesis that both central and peripheral circuits contribute to motor control of the tentacles is supported by the finding that mantle tissues isolated from the VG are capable of crude reflexive responses, but the sensory tentacles become limp and are less responsive than normal [43].

It is likely that the motor control of tentacle extension behaviours in scallops is computationally expensive because many tentacles are involved and each tentacle moves with many degrees of freedom. To address this challenge, we propose that scallops control their array of sensory tentacles in a similar way to how cephalopods control their arms. Like the sensory tentacles of scallops, the arms of cephalopods are hydrostatic organs capable of complex multiaxial movements [44]. Octopus, which have a nervous system composed of roughly 500 million neurons, control their hydrostatic arms using both central ganglionic structures and the medullary cords found in each of their arms [45,46]. Like cephalopods, we propose that scallops control their arrays of sensory tentacles using neural circuits in both their central ganglia and peripheral medullary cords.

## 5. Conclusion: spatial vision in animals with distributed visual systems

Animals with distributed visual systems vary in many ways, including in their body plans, neural structures, locomotory abilities and behavioural repertoires. Such variation suggests these animals may consolidate visual input in different ways and to different degrees. Scallops, for example, display both spatial resolution and spatial vision, but other species with distributed visual systems may display spatial resolution without also displaying spatial vision (i.e. they may have classic 'burglar alarm' visual systems [16]). Further, some distributed visual systems may serve single purposes, whereas others may be associated with multiple light-influenced behaviours [38]. Exploring a wide range of animals with distributed visual systems will help us learn the different and perhaps unexpected ways through which non-cephalized animals convert sensory input into behavioural output.

Data accessibility. Data are available from the Dryad Digital Repository: https://doi.org/10.5061/dryad.gmsbcc2pb [47].

Authors' contributions. D.R.C.: conceptualization, data curation, formal analysis, investigation, methodology, validation, visualization, writing—original draft, writing—review and editing; T.M.H.: data curation, investigation, methodology and writing—original draft; D.I.S.: conceptualization, funding acquisition, investigation, methodology, project administration, resources, supervision, validation, writing—original draft, writing—review and editing. All authors gave final approval for publication and agreed to be held accountable for the work performed therein.

Competing interests. We declare we have no competing interests.

Funding. This research was supported, in part, by IOS award no. 1457148 from the National Science Foundation (to D.I.S.), a Magellan Scholar Award from the University of South Carolina (to T.M.H.) and a Science Undergraduate Research Fellowship from the South Carolina Honors College (to T.M.H.).

Acknowledgements. We thank Lon Wilkens for inspiring this project and Luke Havens for helping to design the behavioural experiments. We also thank Robert Fitak and Sönke Johnsen for feedback on the manuscript and advice on statistical analysis.

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
