## [Peer Review File · Proceedings of the Royal Society B: Biological Sciences]

Review History

RSPB-2020-2550.R0 (Original submission)

Review form: Reviewer 1 (John Kirwan)

Recommendation

Major revision is needed (please make suggestions in comments)

Scientific importance: Is the manuscript an original and important contribution to its field?

Excellent

General interest: Is the paper of sufficient general interest?

Excellent

Quality of the paper: Is the overall quality of the paper suitable?

Good

Is the length of the paper justified?

Yes

Should the paper be seen by a specialist statistical reviewer?

No

Do you have any concerns about statistical analyses in this paper? If so, please specify them explicitly in your report.

Yes

It is a condition of publication that authors make their supporting data, code and materials available - either as supplementary material or hosted in an external repository. Please rate, if applicable, the supporting data on the following criteria.

Is it accessible?

Yes

Is it clear?

Yes

Is it adequate?

Yes

Do you have any ethical concerns with this paper?

No

Comments to the Author

This manuscript concerns an important aspect of vision and its evolution: How does spatial resolution (the ability to detect objects) relate to spatial vision and whether and how spatial vision is achieved in the peculiar scallop visual system? It does so with a behavioral paradigm in which tentacular extension serves as an indicator of visual response.

The method is interesting and the results substantive; however, I have some reservations regarding the statistical analysis which ought to be addressed. This manuscript uses several statistical methods, much of which would benefit from further explanation or justification. Variation between and within individuals is not well accounted for, and some sampling units may be too small. I, therefore, recommend major revisions prior to acceptance, as outlined below.

Major Revisions

Statistical test one - experiments one and two

You have implemented the nonparametric Moore's modified Rayleigh test to test the directedness of the extended tentacles for each individual at all experiments and treatments. You mention that you excluded individuals where none were extended; does that mean that you had none with only one tentacle extended (as this would be untestable)?

Although it is possible to run this test with few replicates, it may not be advisable, even if the effect is generally to be conservative. (Moore notes that the true probability is close to that given by p for $n=30$, $p=0.05$ but more conservative at lower values.) You note that the number of individuals per treatment varies (as some lack extensions) but this concerns me less than the wildly variable numbers of extensions - often few - used to determine significant directedness per individual. Can you justify including those cases with few tentacular extensions? I would be satisfied if this could be shown through simulations or some other suitable method - or if you can evidence that the test is effective even with small sample size. Otherwise, please exclude these instances.

Having condensed the information per individual into p-values, you then counted the number of significant tests from those animals with extended tentacles and observed that this number differs between treatments. Can you justify this as a valid testing procedure? It seems counterintuitive to carry out significance tests per individual and then simply summarize the ratios of significant cases at each treatment. (Especially when variable tentacle sample size means

that the tests are unequally conservative.)

While Moore's modified Rayleigh test has the advantage of considering both angle and extent of the individual tentacles, this nonparametric test constructs ranks based on the vector length, arbitrarily giving equal weight to ordinal differences between extension length, rather than, for instance, weighting directly based on standardized vector length (and so imperfectly preserves this information).

Alternatively, could you, decide on a minimum extension length (possibly based on quantiles from all the data or some other criterion) to consider, below which tentacular extensions are neglected? (From your photographs, there appear to be some tentative tentacular extensions and some much clearer ones.) This would preserve much of the information contained in vector length, while allowing you to use different statistics that avail only of the direction. This would facilitate, for instance, multilevel models considering both individual and tentacle. Modelling according to the von Mises distribution (a symmetrical wrapped normal distribution) seems appropriate and Bayesian mixed models would be robust to very different sample sizes (simply according different uncertainty thereto).

I would be satisfied if you could respond to these concerns and justify the current arrangement.

Statistical test 2, experiment two

234, Table: You note that your negative control better fit a unimodal rather than uniform distribution by the circMLE test (which was not the case for the same treatment in the other experiment), yet you neither explain nor discuss this. How does this impact your other conclusions?

Statistical test 3, experiments one and two.

Did you carried out the Wilcoxon rank sum test on all 16 pairwise combinations of treatments within the two experiments - this is not absolutely clear in the methods (the code seems to require a further datafile). You include a Bonferroni correction in the code which is not mentioned in the text, which is a particularly conservative statistical correction. Yet, here, it is arguably non-conservative, by making rejection of the null (that the number of tentacle extensions do not differ) less likely. Is the only significant difference that between the control and the 6° treatment? How do you interpret this significant result, which you seem to thereafter ignore? I suspect that the number of tentacular extensions is highly variable and only weakly related to the treatments and one significant result is probable when conducting this many significance tests.

Could you not instead carry out a simple (generalized) linear model or analysis of variance? (Perhaps a Poisson regression using lme4 or similar?) This could reasonably look for a monotonic increase of a continuous predictor in experiment two. Such tests can also include random effects to find the variance explained by the individual.

Some comparison between the experiments could be useful to find whether the negative controls or ventral 24° stimulus are comparable - since these ought be equivalent. It would appear to differ for the ventral 24° stimulus.

Statistical test 4

For this method, you do not state on which individuals, tentacles, experiments or treatments the test is carried out, but I infer that you found the circular mean angle and variance per individual at each treatment, as per statistical method 1. I presume since you only mention this in the results in relation to experiment one that you did not carry it out for the second experiment.

You mention calculating z in your methods, but it doesn't appear in the equation directly afterwards (albeit R^* is described by Moore). How did you fit the linear regression (I see from the code you used `lm()` in R, which uses OLS)? Did you test your model fit in any way? Please

describe this method more thoroughly (or at least cite relevant literature).

Can you safely assume that these data ought to be approximately normal (once transformed) and that it is ok to ignore individual effects?

Minor Revisions:

50: degree to which

101,121: The acronym NSW is only used twice and is redundant

68, 131: Is there a reason that you did not use attempt a stimulus size in the range you have cited as the (reported) physiological limit of scallop resolution?

Please more clearly indicate which analysis is which – ideally, with descriptive names for the statistical methods – and on which experiment it is applied.

Do you have an estimate of irradiance (or illuminance) from the set up? I presume that the lights you used are quite bright, even with the diffuser, though it is important that they are.

Review form: Reviewer 2

Recommendation

Major revision is needed (please make suggestions in comments)

Scientific importance: Is the manuscript an original and important contribution to its field?

Excellent

General interest: Is the paper of sufficient general interest?

Good

Quality of the paper: Is the overall quality of the paper suitable?

Good

Is the length of the paper justified?

Yes

Should the paper be seen by a specialist statistical reviewer?

No

Do you have any concerns about statistical analyses in this paper? If so, please specify them explicitly in your report.

No

It is a condition of publication that authors make their supporting data, code and materials available - either as supplementary material or hosted in an external repository. Please rate, if applicable, the supporting data on the following criteria.

Is it accessible?

N/A

Is it clear?

Yes

Is it adequate?

Yes

Do you have any ethical concerns with this paper?

No

Comments to the Author

The manuscript presents interesting findings suggesting that scallops demonstrate spatial vision by directing their sensory tentacles towards visual stimuli at specific points in space. These results are placed within the context of sensory processing in distributed visual systems and attempts to understand how such a system could extract spatial information from the environment to direct behavior. The manuscript is well written and clear. The methods are reasonable and generated some interesting results. However, I am concerned that the ultimate conclusion of the paper, that scallops have true spatial vision including a neural representation of their visual surroundings, is not fully supported by the results to the exclusion of alternate hypotheses. I believe that this conclusion should be tempered considerably, or additional experiments and/or neurological data should be included.

The authors define three possibilities for how the scallop distributed visual system is processing information:

ONE: Spatial resolution: By consolidating information at the level of the entire visual system, an animal can detect objects but not localize them in space. This has been shown previously in scallops by different experiments, is generally thought to be the way many distributed visual systems function. This is useful for escape responses.

TWO: Interocular spatial vision: By consolidating information at the level of each eye and then extracting spatial information by comparison of output between the eyes an animal can locate visual cues in relation to their bodies but not individual eyes.

THREE: Intraocular spatial vision: By comparing visual information between receptors within the same eye an animal can localize cues in relation to both their bodies and eyes.

There is a slim distinction between hypotheses two and three. Behaviorally, the authors suggest that if the scallops preferentially extend tentacles on the side of the animal facing the stimulus AND orient those tentacles towards the stimulus, hypothesis three is confirmed. Either of these outcomes are evidence for true spatial vision. However I have two major concerns that lead to an alternate hypothesis.

1) The authors assume that the distributed eyes are responsible for the tentacle pointing behavior. This would necessitate efferent projections from putative visual processing centers in the PVG to the sensory tentacles. Is there any evidence for the existence of such projections? I am admittedly not familiar with all neurological literature for scallops, so I would happily be corrected on this. Further, is it known conclusively that the tentacles contain no photoreceptors themselves? I do not find it far-fetched, considering the promiscuity of opsin and photoreceptor expression in many animals, that these distributed sensory structures have their own photoreceptive system to some degree. I feel that the authors have extrapolated the high spatial resolution of the distributed eyes shown in previous experiments with the results presented here, leading to the conclusion that true spatial vision is confirmed.

2) The nature of the stimuli in these experiments, black bars on white backgrounds, lends them to being confounded by purely phototactic visual sensitivity. Isoluminant stimuli generated by square, sine-wave, or some other alternation of dark and light components on a grey background could have ameliorated this issue and strengthened the conclusion.

In light of these points, I suggest an additional hypothesis to consider: That the tentacle pointing

behavior could be segregated from the eyes and fall solely under local or distributed phototactic control, not requiring spatial vision.

Scallops display multiple visual behaviors including a startle response, feeding selectivity, tentacle extension, and perhaps orientation during locomotion. In their distributed eyes there are even two segregated photoreceptive systems. Are we certain that there are not additional photoreceptor systems outside of the high-resolution distributed eyes? Scallop brains and non-cephalized, their sensory structures are highly distributed - why not visual behavioral control as well?

One can envision photoreceptors on the apical surface of sensory tentacles, screened by the tentacle tissue, providing them with coarse directional phototactic photoreception. Such receptors could locally detect and orient the tentacles to the presence of dark stimuli by scanning the tentacles over their immediate surroundings. Connections between tentacles in the distributed nervous system, perhaps in the circumpallial nerves, could then promote extension of tentacles in the region of the stimulus and inhibit their extension elsewhere.

To my understanding, this hypothetical system could produce results consistent with those presented here. If I have seriously misunderstood something, or am ignorant of research regarding the nature of the sensory tentacles, I am happy to be corrected. Possible experimental means to remedy these concerns could include repeating these experiments with isoluminant stimuli alongside regional or total ablation of the distributed eyes. However, being unfamiliar with scallops, I am not sure of the feasibility of that manipulation. Also, additional evidence of a circuit connecting the eyes to the sensory tentacle motor control and evidence against local photoreception in the tentacles would confirm the conclusions of this study.

Minor points:

46-62: A clever introductory or supplemental figure would greatly help the broad readership of Proceedings B to understand the competing hypothesis that the experiments endeavor to test. Distributed vision is a difficult concept to grasp even for experienced vision researchers, and it took me several passes of the introduction to diagram out exactly what was being described.

159 - Did the authors test if the cases of non-extension are correlated with any particular treatment before excluding them?

167 - Does CircMLE account for the hinge-limited angles at which a scallop can extend its tentacles? I am admittedly unfamiliar with this package.

189, 220 - equations are not rendered properly in the PDF.

Decision letter (RSPB-2020-2550.R0)

07-Dec-2020

Dear Dr Speiser:

I am writing to inform you that your manuscript RSPB-2020-2550 entitled "Panoramic spatial vision in the bay scallop *Argopecten irradians*" has, in its current form, been rejected for publication in Proceedings B.

This action has been taken on the advice of referees, who have recommended that substantial revisions are necessary. With this in mind we would be happy to consider a resubmission,

provided the comments of the referees are fully addressed. However please note that this is not a provisional acceptance.

Sincerely,
Dr Sasha Dall
mailto:proceedingsb@royalsociety.org

Associate Editor
Board Member: 1
Comments to Author:

Two expert reviewers have now read your manuscript and as you will see both are very positive, and are of the opinion that your findings are potentially both exciting and interesting. However, both have a number of significant reservations about certain aspects of your study that require attention prior to publication. Reviewer 1 is worried about the statistics you have used and Reviewer 2 is concerned that your data do not fully support the ultimate conclusion of your manuscript (that scallops have true spatial vision and a neural representation of their visual world) to the exclusion of alternate hypotheses. These concerns should, of course, be addressed.

Reviewer(s)' Comments to Author:

Referee: 1

Comments to the Author(s)

This manuscript concerns an important aspect of vision and its evolution: How does spatial resolution (the ability to detect objects) relate to spatial vision and whether and how spatial vision is achieved in the peculiar scallop visual system? It does so with a behavioral paradigm in which tentacular extension serves as an indicator of visual response.

The method is interesting and the results substantive; however, I have some reservations regarding the statistical analysis which ought to be addressed. This manuscript uses several statistical methods, much of which and would benefit from further explanation or justification. Variation between and within individuals is not well accounted for, and some sampling units may be too small. I, therefore, recommend major revisions prior to acceptance, as outlined below.

Major Revisions

Statistical test one - experiments one and two

You have implemented the nonparametric Moore's modified Rayleigh test to test the directedness of the extended tentacles for each individual at all experiments and treatments. You mention that you excluded individuals where none were extended; does that mean that you had none with only one tentacle extended (as this would be untestable)?

Although it is possible to run this test with few replicates, it may not be advisable, even if the effect is generally to be conservative. (Moore notes that the true probability is close to that given by p for $n=30$, $p=0.05$ but more conservative at lower values.) You note that the number of individuals per treatment varies (as some lack extensions) but this concerns me less than the wildly variable numbers of extensions – often few – used to determine significant directedness per individual. Can you justify including those cases with few tentacular extensions? I would be satisfied if this could be shown through simulations or some other suitable method – or if you can evidence that the test is effective even with small sample size. Otherwise, please exclude these instances.

Having condensed the information per individual into p-values, you then counted the number of significant tests from those animals with extended tentacles and observed that this number differs between treatments. Can you justify this as a valid testing procedure? It seems counterintuitive to carry out significance tests per individual and then simply summarize the ratios of significant cases at each treatment. (Especially when variable tentacle sample size means that the tests are unequally conservative.)

While Moore's modified Rayleigh test has the advantage of considering both angle and extent of the individual tentacles, this nonparametric test constructs ranks based on the vector length, arbitrarily giving equal weight to ordinal differences between extension length, rather than, for instance, weighting directly based on standardized vector length (and so imperfectly preserves this information).

Alternatively, could you, decide on a minimum extension length (possibly based on quantiles from all the data or some other criterion) to consider, below which tentacular extensions are neglected? (From your photographs, there appear to be some tentative tentacular extensions and some much clearer ones.) This would preserve much of the information contained in vector length, while allowing you to use different statistics that avail only of the direction. This would facilitate, for instance, multilevel models considering both individual and tentacle. Modelling according to the von Mises distribution (a symmetrical wrapped normal distribution) seems appropriate and Bayesian mixed models would be robust to very different sample sizes (simply according different uncertainty thereto).

I would be satisfied if you could respond to these concerns and justify the current arrangement.

Statistical test 2, experiment two

234, Table: You note that your negative control better fit a unimodal rather than uniform distribution by the circMLE test (which was not the case for the same treatment in the other experiment), yet you neither explain nor discuss this. How does this impact your other conclusions?

Statistical test 3, experiments one and two.

Did you carried out the Wilcoxon rank sum test on all 16 pairwise combinations of treatments within the two experiments - this is not absolutely clear in the methods (the code seems to require a further datafile). You include a Bonferroni correction in the code which is not mentioned in the text, which is a particularly conservative statistical correction. Yet, here, it is arguably non-conservative, by making rejection of the null (that the number of tentacle extensions do not differ) less likely. Is the only significant difference that between the control and the 6° treatment? How do you interpret this significant result, which you seem to thereafter ignore? I suspect that the

number of tentacular extensions is highly variable and only weakly related to the treatments and one significant result is probable when conducting this many significance tests.

Could you not instead carry out a simple (generalized) linear model or analysis of variance? (Perhaps a Poisson regression using lme4 or similar?) This could reasonably look for a monotonic increase of a continuous predictor in experiment two. Such tests can also include random effects to find the variance explained by the individual.

Some comparison between the experiments could be useful to find whether the negative controls or ventral 24° stimulus are comparable – since these ought to be equivalent. It would appear to differ for the ventral 24° stimulus.

Statistical test 4

For this method, you do not state on which individuals, tentacles, experiments or treatments the test is carried out, but I infer that you found the circular mean angle and variance per individual at each treatment, as per statistical method 1. I presume since you only mention this in the results in relation to experiment one that you did not carry it out for the second experiment.

You mention calculating z in your methods, but it doesn't appear in the equation directly afterwards (albeit R^* is described by Moore). How did you fit the linear regression (I see from the code you used `lm()` in R, which uses OLS)? Did you test your model fit in any way? Please describe this method more thoroughly (or at least cite relevant literature).

Can you safely assume that these data ought to be approximately normal (once transformed) and that it is ok to ignore individual effects?

Minor Revisions:

50: degree to which

101,121: The acronym NSW is only used twice and is redundant

68, 131: Is there a reason that you did not use attempt a stimulus size in the range you have cited as the (reported) physiological limit of scallop resolution?

Please more clearly indicate which analysis is which – ideally, with descriptive names for the statistical methods – and on which experiment it is applied.

Do you have an estimate of irradiance (or illuminance) from the set up? I presume that the lights you used are quite bright, even with the diffuser, though it is important that they are.

Referee: 2

Comments to the Author(s)

The manuscript presents interesting findings suggesting that scallops demonstrate spatial vision by directing their sensory tentacles towards visual stimuli at specific points in space. These results are placed within the context of sensory processing in distributed visual systems and attempts to understand how such a system could extract spatial information from the environment to direct behavior. The manuscript is well written and clear. The methods are reasonable and generated some interesting results. However, I am concerned that the ultimate conclusion of the paper, that scallops have true spatial vision including a neural representation of their visual surroundings, is not fully supported by the results to the exclusion of alternate hypotheses. I believe that this conclusion should be tempered considerably, or additional experiments and/or neurological data should be included.

The authors define three possibilities for how the scallop distributed visual system is processing information:

ONE: Spatial resolution: By consolidating information at the level of the entire visual system, an animal can detect objects but not localize them in space. This has been shown previously in scallops by different experiments, is generally thought to be the way many distributed visual systems function. This is useful for escape responses.

TWO: Interocular spatial vision: By consolidating information at the level of each eye and then extracting spatial information by comparison of output between the eyes an animal can locate visual cues in relation to their bodies but not individual eyes.

THREE: Intraocular spatial vision: By comparing visual information between receptors within the same eye an animal can localize cues in relation to both their bodies and eyes.

There is a slim distinction between hypotheses two and three. Behaviorally, the authors suggest that if the scallops preferentially extend tentacles on the side of the animal facing the stimulus AND orient those tentacles towards the stimulus, hypothesis three is confirmed. Either of these outcomes are evidence for true spatial vision. However I have two major concerns that lead to an alternate hypothesis.

1) The authors assume that the distributed eyes are responsible for the tentacle pointing behavior. This would necessitate efferent projections from putative visual processing centers in the PVG to the sensory tentacles. Is there any evidence for the existence of such projections? I am admittedly not familiar with all neurological literature for scallops, so I would happily be corrected on this. Further, is it known conclusively that the tentacles contain no photoreceptors themselves? I do not find it far-fetched, considering the promiscuity of opsin and photoreceptor expression in many animals, that these distributed sensory structures have their own photoreceptive system to some degree. I feel that the authors have extrapolated the high spatial resolution of the distributed eyes shown in previous experiments with the results presented here, leading to the conclusion that true spatial vision is confirmed.

2) The nature of the stimuli in these experiments, black bars on white backgrounds, lends them to being confounded by purely phototactic visual sensitivity. Isoluminant stimuli generated by square, sine-wave, or some other alternation of dark and light components on a grey background could have ameliorated this issue and strengthened the conclusion.

In light of these points, I suggest an additional hypothesis to consider: That the tentacle pointing behavior could be segregated from the eyes and fall solely under local or distributed phototactic control, not requiring spatial vision.

Scallops display multiple visual behaviors including a startle response, feeding selectivity, tentacle extension, and perhaps orientation during locomotion. In their distributed eyes there are even two segregated photoreceptive systems. Are we certain that there are not additional photoreceptor systems outside of the high-resolution distributed eyes? Scallop brains and non-cephalized, their sensory structures are highly distributed - why not visual behavioral control as well?

One can envision photoreceptors on the apical surface of sensory tentacles, screened by the tentacle tissue, providing them with coarse directional phototactic photoreception. Such receptors could locally detect and orient the tentacles to the presence of dark stimuli by scanning the tentacles over their immediate surroundings. Connections between tentacles in the distributed nervous system, perhaps in the circumpallial nerves, could then promote extension of tentacles in the region of the stimulus and inhibit their extension elsewhere.

To my understanding, this hypothetical system could produce results consistent with those presented here. If I have seriously misunderstood something, or am ignorant of research regarding the nature of the sensory tentacles, I am happy to be corrected. Possible experimental means to remedy these concerns could include repeating these experiments with isoluminant

stimuli alongside regional or total ablation of the distributed eyes. However, being unfamiliar with scallops, I am not sure of the feasibility of that manipulation. Also, additional evidence of a circuit connecting the eyes to the sensory tentacle motor control and evidence against local photoreception in the tentacles would confirm the conclusions of this study.

Minor points:

46-62: A clever introductory or supplemental figure would greatly help the broad readership of Proceedings B to understand the competing hypothesis that the experiments endeavor to test. Distributed vision is a difficult concept to grasp even for experienced vision researchers, and it took me several passes of the introduction to diagram out exactly what was being described.

159 - Did the authors test if the cases of non-extension are correlated with any particular treatment before excluding them?

167 - Does CircMLE account for the hinge-limited angles at which a scallop can extend its tentacles? I am admittedly unfamiliar with this package.

189, 220 - equations are not rendered properly in the PDF.

Author's Response to Decision Letter for (RSPB-2020-2550.R0)

See Appendix A.

Decision letter (RSPB-2021-1730.R0)

08-Oct-2021

Dear Dr Speiser

I am pleased to inform you that your manuscript RSPB-2021-1730 entitled "Panoramic spatial vision in the bay scallop *Argopecten irradians*" has been accepted for publication in Proceedings B.

The referee(s) have recommended publication, but also suggest some minor revisions to your manuscript. Therefore, I invite you to respond to the referee(s)' comments and revise your manuscript. Because the schedule for publication is very tight, it is a condition of publication that you submit the revised version of your manuscript within 7 days. If you do not think you will be able to meet this date please let us know.

When submitting your revised manuscript, you will be able to respond to the comments made by the referee(s) and upload a file "Response to Referees". You can use this to document any changes you make to the original manuscript. We require a copy of the manuscript with revisions made

since the previous version marked as 'tracked changes' to be included in the 'response to referees' document.

Sincerely,
Dr Sasha Dall
mailto: proceedingsb@royalsociety.org

Associate Editor
Board Member
Comments to Author:

As you will see, both reviewers are very happy with the efforts you have made to address their comments, but they still have a small number of minor suggestions that should be addressed prior to publication.

Reviewer comments:

Reviewer 1

I appreciate the hard work by the authors to address the points raised by myself and the other reviewer. The manuscript is greatly improved, and I find the new experiments especially compelling. There is now much better evidence that this interesting behaviour of tentacle-pointing at visual stimuli is indicative of spatial vision and that this behaviour is most likely being mediated by the mantle eyes. I now think the manuscript is acceptable pending a few minor revisions and clarifications. This remains a very surprising finding that requires further investigation to understand more clearly.

1) The new optomotor experiments clearly show the tentacles extending towards, and apparently tracking, an isoluminant bar stimulus. The response is surprisingly impressive! I am now much more convinced that the tentacle extension is likely driven by the mantle eyes rather than some hypothetical photoreceptors in the tentacles. I would not totally rule out some sort of photoreceptors in these tentacles playing some role in this behavior. The lack of obvious pigment cup structures does not necessarily rule out the presence of photoreceptors, and the photoreceptors could still be screened to some extent by the tissue of the tentacle. Regardless, the resolving power needed to detect the isoluminant bars would be difficult to achieve without the eyes.

2) I appreciate the added detail regarding the innervation and responses of the tentacles. Considering the new data, and if the eyes indeed project only to the central ganglia, there must be some neural representation of visual space in the central ganglia that is then used to direct the tentacle pointing.

3) Was any attempt made to repeat the original experiments with static isoluminant bar stimuli? It is a little bit surprising that such an approach was not reported, and the experiment was changed to an optomotor approach. If the experimenters found that motion was necessary to elicit a tentacle response to isoluminant stimuli, that could be useful to understanding the nature of this behavioural response and should be reported. If static isoluminant bars were not attempted, why not? My main remaining reservation is that the eyes could somehow provide coarse intraocular spatial vision for directing tentacle extension and then local photoreceptors in the tentacles could be assisting in orientation of those tentacles towards the stimulus. The rotating isoluminant stimulus does not seem to test this as clearly and would have perhaps been better served with static isoluminant bar stimuli.

4) I still think a diagram explaining spatial resolution versus with inter/intra-ocular spatial vision would greatly help with broad understandability, but I will leave that at the discretion of the editor.

5) There are still some issues with formatting of Greek letters in the text, but that is easily remedied.

Reviewer 2

I thank the authors for addressing the comments in the prior round of review and for adding another substantial experiment. My major concerns have been satisfied by these comments and additions. I include below some comments below, which the authors should consider prior to publication.

Comments

In your reply to reviewer comments, you state that you exclude from analysis trials in which scallops extend fewer than three tentacles. I cannot find this in the methods. Have I missed something? If not, this must be included.

L. 113. Do you intend to list the manufacturer or the supplier? Adafruit do not make Arduino. I still dislike the practice of comparing proportions of rejected null hypotheses of undirectedness, as applied to individuals at each treatment. It seems incongruous to apply severe testing in the form of null hypothesis decision criteria and then to be satisfied with merely summarizing the proportions of those very tests. However, the effect size is large and this method does not apparently engender a systemic bias, so it is not problematic in this context.

Author's Response to Decision Letter for (RSPB-2021-1730.R0)

See Appendix B.

Decision letter (RSPB-2021-1730.R1)

20-Oct-2021

Dear Mr Chappell

I am pleased to inform you that your manuscript entitled "Panoramic spatial vision in the bay scallop *Argopecten irradians*" has been accepted for publication in Proceedings B.

Data Accessibility section

Open Access

Paper charges

Sincerely,

Appendix A

Response to Reviews on Manuscript ID RSPB-2020-2550

Associate Editor Comments to Author:

Two expert reviewers have now read your manuscript and as you will see both are very positive, and are of the opinion that your findings are potentially both exciting and interesting. However, both have a number of significant reservations about certain aspects of your study that require attention prior to publication. Reviewer 1 is worried about the statistics you have used and Reviewer 2 is concerned that your data do not fully support the ultimate conclusion of your manuscript (that scallops have true spatial vision and a neural representation of their visual world) to the exclusion of alternate hypotheses.

RESPONSE: We appreciate the reviewers' enthusiasm about our work. We have responded to each point raised by the referees below.

Referee 1 Comments to the Author(s):

This manuscript concerns an important aspect of vision and its evolution: How does spatial resolution (the ability to detect objects) relate to spatial vision and whether and how spatial vision is achieved in the peculiar scallop visual system? It does so with a behavioral paradigm in which tentacular extension serves as an indicator of visual response. The method is interesting and the results substantive; however, I have some reservations regarding the statistical analysis which ought to be addressed. This manuscript uses several statistical methods, much of which and would benefit from further explanation or justification. Variation between and within individuals is not well accounted for, and some sampling units may be too small. I, therefore, recommend major revisions prior to acceptance, as outlined below.

RESPONSE: We thank Reviewer 1 for their thoughtful and thorough review. We made changes to our statistical methodology where appropriate (see details below). To our initial experiments, we added a new experiment on the responses of scallops to rotating visual stimuli. This new experiment is a substantial addition to our project: for it, we ran 80 additional trials and analyzed 10,400 video frames. The results of the new experiment support our conclusion that scallops have spatial vision. For the trials with static visual stimuli, we developed a quantitative methodology appropriate for determining the directedness of the population of tentacles extended by an individual scallop. An advantage of this approach is that we were able to apply it iteratively to analyze the directedness of tentacles extended by scallops in response to rotating visual stimuli.

Major Revisions

Statistical test one - experiments one and two

You have implemented the nonparametric Moore's modified Rayleigh test to test the directedness of the extended tentacles for each individual at all experiments and treatments. You mention that you excluded individuals where none were extended; does that mean that you had none with only one tentacle extended (as this would be untestable)?

RESPONSE: We excluded from our analyses trials in which scallops extended fewer than 3 tentacles, so we did not analyze trials in which scallops extended only one tentacle.

Although it is possible to run this test with few replicates, it may not be advisable, even if the effect is generally to be conservative. (Moore notes that the true probability is close to that given by p for $n=30$, $p=0.05$ but more conservative at lower values.) You note that the number of individuals per treatment varies (as some lack extensions) but this concerns me less than the wildly variable numbers of extensions – often few - used to determine significant directedness per individual. Can you justify including those cases with few tentacular extensions? I would be satisfied if this could be shown through simulations or some other suitable method – or if you can evidence that the test is effective even with small sample size. Otherwise, please exclude these instances.

RESPONSE: We emphasize that the goal of this analysis is to quantify the directed behavior of individual scallops. Scallops can choose how many tentacles they extend and how they direct those tentacles. Any analysis of the tentacle extension behaviors of scallops should include the numbers of tentacles extended by animals and where those tentacles are pointing. From our observations, scallops sometimes choose to show directed responses with only a few tentacles, and we do not want to omit these directed responses from our analysis because they represent deliberate and potentially meaningful responses by animals.

Having condensed the information per individual into p-values, you then counted the number of significant tests from those animals with extended tentacles and observed that this number differs between treatments. Can you justify this as a valid testing procedure? It seems counterintuitive to carry out significance tests per individual and then simply summarize the ratios of significant cases at each treatment. (Especially when variable tentacle sample size means that the tests are unequally conservative.)

RESPONSE: Scallops tend to extend tentacles even in negative control conditions (no visual stimuli present), but these extended tentacles are not directed or are weakly directed. Tentacle extension behavior represents a scallop's focused attention, so when a scallop chooses to extend only a few tentacles in one direction it is simultaneously choosing to retract the rest of its tentacle array.

Ideally, we could monitor and quantify the entirety of the tentacle array at all times, but tentacles cannot be tracked if they do not extend beyond the shell margins. In short, although analyzing a relatively small number of extended tentacles seems like liberal testing, there are numerous retracted tentacles making this a more conservative measure of directedness. We realize the difficulties in comparing the directedness of scallops which have chosen to extend different numbers of tentacles, which is why we chose to compare proportions of scallops that showed significantly directed arrays of tentacles.

While Moore's modified Rayleigh test has the advantage of considering both angle and extent of the individual tentacles, this nonparametric test constructs ranks based on the vector length, arbitrarily giving equal weight to ordinal differences between extension length, rather than, for instance, weighting directly based on standardized vector length (and so imperfectly preserves this information).

Alternatively, could you, decide on a minimum extension length (possibly based on quantiles from all the data or some other criterion) to consider, below which tentacular extensions are neglected? (From your photographs, there appear to be some tentative tentacular extensions and some much clearer ones.) This would preserve much of the information contained in vector length, while allowing you to use different statistics that avail only of the direction. This would facilitate, for instance, multilevel models considering both individual and tentacle. Modelling according to the von Mises distribution (a symmetrical wrapped normal distribution) seems appropriate and Bayesian mixed models would be robust to very different sample sizes (simply according different uncertainty thereto). I would be satisfied if you could respond to these concerns and justify the current arrangement.

RESPONSE: Difficulties in analyzing tentacle extension behaviors in scallops include the tentacles having different lengths and having patchy distributions. In our video analysis, we utilized the longitudinal tracking of individual tentacles to get an average tentacle length over the course of a trial, and from this calculate a relative tentacle length for each frame. We understand the limitations of the static trials, but we believe analyzing both tentacle length and direction is important to quantitatively describe how scallops respond to visual stimuli. As we note above, scallops can choose the lengths to which they extend their tentacles and the directions in which they point their tentacles, so it is appropriate to include both factors in our analyses.

Statistical test 2, experiment two

234, Table: You note that your negative control better fit a unimodal rather than uniform distribution by the circMLE test (which was not the case for the same treatment in the other experiment), yet you neither explain nor discuss this. How does this impact your other conclusions?

RESPONSE: Even in the absence of visual stimuli, scallops extend tentacles to monitor and explore their environment. Because the tentacles are not distributed evenly across the mantle tissue, these responses may appear unimodal or weakly directed in the absence of stimuli. This makes it important to look at both the magnitude and direction of the distribution of directedness in the analysis of tentacle extension behavior. As described in our methods, we first looked at the magnitude of the directedness vector (R^*) and then separately analyzed the distribution of direction of the directedness vector (ϕ^*). For the negative control, we first note that the proportion of significant directed responses is low (1/19), and then note that the distribution of directions of the directedness vectors is unimodal. If these analyses are interpreted in concert, it demonstrates that scallops in the negative control group showed weakly directed responses that were distributed unimodally. This finding does not impact our other conclusions because scallops from non-control treatments showed more strongly directed responses. Further, the results from our new trials with rotating visual stimuli help support our claim that scallops responded to visual stimuli differently than they responded to the negative controls.

Statistical test 3, experiments one and two.

Did you carry out the Wilcoxon rank sum test on all 16 pairwise combinations of treatments within the two experiments - this is not absolutely clear in the methods (the code seems to require a further datafile). You include a Bonferroni correction in the code which is not mentioned in the text, which is a particularly conservative statistical correction. Yet, here, it is arguably non-conservative, by making rejection of the null (that the number of tentacle extensions do not differ) less likely. Is the only significant difference that between the control and the 6° treatment? How do you interpret this significant result, which you seem to thereafter ignore? I suspect that the number of tentacular extensions is highly variable and only weakly related to the treatments and one significant result is probable when conducting this many significance tests.

Could you not instead carry out a simple (generalized) linear model or analysis of variance? (Perhaps a Poisson regression using lme4 or similar?) This could reasonably look for a monotonic increase of a continuous predictor in experiment two. Such tests can also include random effects to find the variance explained by the individual. Some comparison between the experiments could be useful to find whether the negative controls or ventral 24° stimulus are comparable – since these ought to be equivalent. It would appear to differ for the ventral 24° stimulus.

RESPONSE: We replaced the pairwise tests with two (one for each experiment) omnibus Friedman rank sum tests of the form: (extended tentacle count) ~ (treatment) | (scallop identity). From the results of these tests, we found that extended tentacle count did not have a significant relationship with the treatments. This supports our conclusion that the numbers of tentacles extended

by scallops did not depend on the presence, size, or location of visual stimuli. We have edited the text accordingly to accommodate these additions and alterations.

Statistical test 4

For this method, you do not state on which individuals, tentacles, experiments or treatments the test is carried out, but I infer that you found the circular mean angle and variance per individual at each treatment, as per statistical method 1. I presume since you only mention this in the results in relation to experiment one that you did not carry it out for the second experiment.

You mention calculating z in your methods, but it doesn't appear in the equation directly afterwards (albeit R^* is described by Moore). How did you fit the linear regression (I see from the code you used `lm()` in R, which uses OLS)? Did you test your model fit in any way? Please describe this method more thoroughly (or at least cite relevant literature). Can you safely assume that these data ought to be approximately normal (once transformed) and that it is ok to ignore individual effects?

RESPONSE: In our revised manuscript we provide more detail about our linear model and altered the linear model to address Reviewer 1's concerns about this analysis. We altered the linear model to include the z parameter (linear distance). The linear model is now of the form: $R^* = 1 + \beta_1 + \beta_2(z)$. We fit the linear regression using ordinary least squares fitting. We tested model fit by calculating the R^2 value. We assessed the normality of the residuals of the model using a Shapiro Wilks test which yielded a non-significant p -value ($P > 0.05$).

Minor Revisions:

50: degree to which

RESPONSE: We have made this edit.

101,121: The acronym NSW is only used twice and is redundant

RESPONSE: We denote this acronym at its first usage and now use it five times.

68, 131: Is there a reason that you did not use attempt a stimulus size in the range you have cited as the (reported) physiological limit of scallop resolution?

RESPONSE: Individual eyes from scallops have a resolution limit of about 2 degrees. However, the resolution limit of the many-eyed scallop visual system remains unclear. In response to this query we used isoluminant visual stimuli with angular widths of 2, 5, 10, and 20 degrees in the new experiment we added to our revised manuscript. In this new experiment, scallops responded to visual stimuli

with angular widths as narrow as 5 degrees. They did not respond to the 2 degree visual stimulus or the negative control.

Please more clearly indicate which analysis is which – ideally, with descriptive names for the statistical methods – and on which experiment it is applied.

RESPONSE: We have made edits where appropriate to help alleviate confusion. In our manuscript, we often describe which data sets the different parameters are extracted from. In subsequent analyses on these parameters, we have not repeated a description of the parameter to remove redundancy.

Do you have an estimate of irradiance (or illuminance) from the set up? I presume that the lights you used are quite bright, even with the diffuser, though it is important that they are.

RESPONSE: The approximate downwelling irradiance experienced by the scallops in the static trials was $\sim 2 \times 10^{15}$ photons $\text{cm}^{-2} \text{s}^{-1}$ ($\sim 1,000$ lux, equivalent to downwelling irradiance on an overcast day) The dynamic trials were lit from above and below, making it challenging to give a total irradiance value (we did not have an integrating sphere on hand). The total irradiance in dynamic trials was not more than 4×10^{15} photons $\text{cm}^{-2} \text{s}^{-1}$. These conditions were not unnaturally bright for bay scallops, which tend to live in shallow inshore habitats.

Referee 2 Comments to the Author(s):

The manuscript presents interesting findings suggesting that scallops demonstrate spatial vision by directing their sensory tentacles towards visual stimuli at specific points in space. These results are placed within the context of sensory processing in distributed visual systems and attempts to understand how such a system could extract spatial information from the environment to direct behavior. The manuscript is well written and clear. The methods are reasonable and generated some interesting results. However, I am concerned that the ultimate conclusion of the paper, that scallops have true spatial vision including a neural representation of their visual surroundings, is not fully supported by the results to the exclusion of alternate hypotheses. I believe that this conclusion should be tempered considerably, or additional experiments and/or neurological data should be included.

RESPONSE: We thank Reviewer 2 for their thoughtful and thorough review. We have performed additional behavioral trials using rotating isoluminant stimuli that address Reviewer 2's concerns and bolster the conclusion of our paper that scallops have both spatial resolution and spatial vision.

The authors define three possibilities for how the scallop distributed visual system is processing information:

ONE: Spatial resolution: By consolidating information at the level of the entire visual system, an animal can detect objects but not localize them in space. This has been shown previously in scallops by different experiments, is generally thought to be the way many distributed visual systems function. This is useful for escape responses.

TWO: Interocular spatial vision: By consolidating information at the level of each eye and then extracting spatial information by comparison of output between the eyes an animal can locate visual cues in relation to their bodies but not individual eyes.

THREE: Intraocular spatial vision: By comparing visual information between receptors within the same eye an animal can localize cues in relation to both their bodies and eyes.

There is a slim distinction between hypotheses two and three. Behaviorally, the authors suggest that if the scallops preferentially extend tentacles on the side of the animal facing the stimulus AND orient those tentacles towards the stimulus, hypothesis three is confirmed. Either of these outcomes are evidence for true spatial vision. However I have two major concerns that lead to an alternate hypothesis.

RESPONSE: The distinction between hypotheses two and three is not slim, at least for some visual systems. In scallops, for example, interocular spatial vision would likely result in a coarse map of the visual world that contained mere dozens of pixels. In contrast, intraocular spatial vision could result in a fine visual map containing many thousands of pixels. Scallops, of course, may have spatial vision that falls somewhere between these two categories. Nevertheless, presenting these categories gives us a conceptual framework from which we can generate testable hypotheses.

1) The authors assume that the distributed eyes are responsible for the tentacle pointing behavior. This would necessitate efferent projections from putative visual processing centers in the PVG to the sensory tentacles. Is there any evidence for the existence of such projections? I am admittedly not familiar with all neurological literature for scallops, so I would happily be corrected on this. Further, is it known conclusively that the tentacles contain no photoreceptors themselves? I do not find it far-fetched, considering the promiscuity of opsin and photoreceptor expression in many animals, that these distributed sensory structures have their own photoreceptive system to some degree. I feel that the authors have extrapolated the high spatial resolution of the distributed eyes shown in previous experiments with the results presented here, leading to the conclusion that true spatial vision is confirmed.

RESPONSE: First, there are efferent projections from the visual processing centers of scallops (the lateral lobes of the visceral ganglia) back to the sensory tentacles. We have made this point more strongly in the discussion section of our

text. It is possible that the sensory tentacles contain photoreceptors, but the tentacles are narrow and translucent and close examination does not indicate the presence of any eyespot-like structures. From this, it seems highly unlikely that the sensory tentacles themselves can gather spatial information about light.

2) The nature of the stimuli in these experiments, black bars on white backgrounds, lends them to being confounded by purely phototactic visual sensitivity. Isoluminant stimuli generated by square, sine-wave, or some other alternation of dark and light components on a grey background could have ameliorated this issue and strengthened the conclusion.

RESPONSE: The reviewer raises an important point here. Accordingly, we used isoluminant stimuli in the new experiment we added to our manuscript. The results from our new experiment are consistent with the results from our initial experiments, strengthening our conclusion that the tentacle extension behaviors of scallops are indeed guided by spatial information about light.

In light of these points, I suggest an additional hypothesis to consider: That the tentacle pointing behavior could be segregated from the eyes and fall solely under local or distributed phototactic control, not requiring spatial vision.

Scallops display multiple visual behaviors including a startle response, feeding selectivity, tentacle extension, and perhaps orientation during locomotion. In their distributed eyes there are even two segregated photoreceptive systems. Are we certain that there are not additional photoreceptor systems outside of the high-resolution distributed eyes? Scallop brains and non-cephalized, their sensory structures are highly distributed - why not visual behavioral control as well?

One can envision photoreceptors on the apical surface of sensory tentacles, screened by the tentacle tissue, providing them with coarse directional phototactic photoreception. Such receptors could locally detect and orient the tentacles to the presence of dark stimuli by scanning the tentacles over their immediate surroundings. Connections between tentacles in the distributed nervous system, perhaps in the circumpallial nerves, could then promote extension of tentacles in the region of the stimulus and inhibit their extension elsewhere.

To my understanding, this hypothetical system could produce results consistent with those presented here. If I have seriously misunderstood something, or am ignorant of research regarding the nature of the sensory tentacles, I am happy to be corrected. Possible experimental means to remedy these concerns could include repeating these experiments with isoluminant stimuli alongside regional or total ablation of the distributed eyes. However, being unfamiliar with scallops, I am not sure of the feasibility of that manipulation. Also, additional evidence of a circuit connecting the eyes to the sensory tentacle motor control and evidence against local photoreception in the tentacles would confirm the conclusions of this study.

RESPONSE: More than half the scallops we tested (9 out of 16) tracked rotating isoluminant visual stimuli with angular widths as narrow as 5 degrees (see Figure 4c). In comparison, none of the scallops we tested followed the negative control. It would be unprecedented (and we think implausible) for an animal to demonstrate this level of spatial vision using dispersed direction-sensitive photoreceptors. We have also added text that discusses previous electrophysiological experiments in scallops in which sensory tentacles on mantle tissue isolated from the visceral ganglion were found to become limp and weakly responsive. These experiments indicate that the both central and peripheral neural circuits likely contribute to the control of the sensory tentacles.

Minor points:

46-62: A clever introductory or supplemental figure would greatly help the broad readership of Proceedings B to understand the competing hypothesis that the experiments endeavor to test. Distributed vision is a difficult concept to grasp even for experienced vision researchers, and it took me several passes of the introduction to diagram out exactly what was being described.

RESPONSE: We have reviewed and edited the text of the Introduction section. We hope we have been able to clarify our hypotheses.

159 - Did the authors test if the cases of non-extension are correlated with any particular treatment before excluding them?

RESPONSE: We checked if these cases were correlated with treatment prior to exclusion and found that they were not significantly correlated with any particular treatment.

167 - Does CircMLE account for the hinge-limited angles at which a scallop can extend its tentacles? I am admittedly unfamiliar with this package.

RESPONSE: We acknowledge that scallops do not have an idealized 360 degree array of homogeneously arranged tentacles. Scallops are able to extend tentacles around their dorsal hinge. We used tentacle angle in our analysis which accommodates hinge-adjacent tentacles pointing around the hinge. In addition, we have openly acknowledged that the non-uniform circular distribution of tentacles likely leads to some bias in our analysis which is an unavoidable aspect of scallop morphology.

189, 220 - equations are not rendered properly in the PDF.

RESPONSE: We identified and fixed this problem.

Appendix B

Response to Reviews on Manuscript ID RSPB-2021-1730

Associate Editor
Board Member
Comments to Author:

As you will see, both reviewers are very happy with the efforts you have made to address their comments, but they still have a small number of minor suggestions that should be addressed prior to publication.

RESPONSE: We appreciate the reviewers' enthusiasm about our work. We have responded to each point raised by the reviewers below.

Reviewer comments:

REVIEWER 1

I appreciate the hard work by the authors to address the points raised by myself and the other reviewer. The manuscript is greatly improved, and I find the new experiments especially compelling. There is now much better evidence that this interesting behaviour of tentacle-pointing at visual stimuli is indicative of spatial vision and that this behaviour is most likely being mediated by the mantle eyes. I now think the manuscript is acceptable pending a few minor revisions and clarifications. This remains a very surprising finding that requires further investigation to understand more clearly.

RESPONSE: We thank Reviewer 1 for their thoughtful review and for recognizing our efforts to bolster our results with additional behavioral trials and analyses.

1) The new optomotor experiments clearly show the tentacles extending towards, and apparently tracking, an isoluminant bar stimulus. The response is surprisingly impressive! I am now much more convinced that the tentacle extension is likely driven by the mantle eyes rather than some hypothetical photoreceptors in the tentacles. I would not totally rule out some sort of photoreceptors in these tentacles playing some role in this behavior. The lack of obvious pigment cup structures does not necessarily rule out the presence of photoreceptors, and the photoreceptors could still be screened to some extent by the tissue of the tentacle. Regardless, the resolving power needed to detect the isoluminant bars would be difficult to achieve without the eyes.

RESPONSE: We do not deny that photoreceptors may be present in the tentacles. These hypothetical tentacle photoreceptors could be screened to some extent by tissue, but the tentacles are narrow and translucent. We therefore consider it highly unlikely that a tentacle-based photoreceptor array would have the resolving power required to detect the narrow isoluminant bars we used as visual stimuli in our trials.

2) I appreciate the added detail regarding the innervation and responses of the tentacles. Considering the new data, and if the eyes indeed project only to the central ganglia, there must be some neural representation of visual space in the central ganglia that is then used to direct the tentacle pointing.

RESPONSE: We share Reviewer 1's enthusiasm for the implications our findings hold for the visual processing circuits of scallops. We plan to publish on this topic soon.

3) Was any attempt made to repeat the original experiments with static isoluminant bar stimuli? It is a little bit surprising that such an approach was not reported, and the experiment was changed to an optomotor approach. If the experimenters found that motion was necessary to elicit a tentacle response to isoluminant stimuli, that could be useful to understanding the nature of this behavioural response and should be reported. If static isoluminant bars were not attempted, why not?

RESPONSE: We did not repeat the original experiments with static isoluminant bar stimuli. We could have repeated the static trials using an isoluminant bar stimuli, but we decided that the dynamic tracking of visual stimuli was a more robust demonstration of spatial vision by scallops. From preliminary observations, we find that scallops are as likely to extend their sensory tentacles towards static isoluminant static bar stimuli as they are towards non-isoluminant static bar stimuli.

My main remaining reservation is that the eyes could somehow provide coarse intraocular spatial vision for directing tentacle extension and then local photoreceptors in the tentacles could be assisting in orientation of those tentacles towards the stimulus. The rotating isoluminant stimulus does not seem to test this as clearly and would have perhaps been better served with static isoluminant bar stimuli.

RESPONSE: Repeating our response from above, we do not deny the possibility that photoreceptors are present in the tentacles. These hypothetical tentacle photoreceptors could be screened to some extent by tissue, but the tentacles are narrow and translucent. We therefore consider it highly unlikely that a tentacle-based photoreceptor array would have the resolving power required to detect the narrow isoluminant bars we used as visual stimuli in our trials.

4) I still think a diagram explaining spatial resolution versus with inter/intra-ocular spatial vision would greatly help with broad understandability, but I will leave that at the discretion of the editor.

RESPONSE: Our manuscript currently contains the maximum number of figures allowed by *Proc B*. We will add an additional figure if requested to do so by the Editor (in this scenario we would like to talk with the Editor about which of our figures should become supplemental or how we should cover the costs associated with adding another figure).

5) There are still some issues with formatting of Greek letters in the text, but that is easily remedied.

RESPONSE: We have corrected these issues.

REVIEWER 2

I thank the authors for addressing the comments in the prior round of review and for adding another substantial experiment. My major concerns have been satisfied by these comments and additions. I include below some comments below, which the authors should consider prior to publication.

RESPONSE: Thanks!

In your reply to reviewer comments, you state that you exclude from analysis trials in which scallops extend fewer than three tentacles. I cannot find this in the methods. Have I missed something? If not, this must be included.

RESPONSE: We have edited the manuscript to include this information (see lines 166-167 and 227-228).

L. 113. Do you intend to list the manufacturer or the supplier? Adafruit do not make Arduino.

RESPONSE: We have edited the manuscript to include this information (see lines 113-114).

I still dislike the practice of comparing proportions of rejected null hypotheses of undirectedness, as applied to individuals at each treatment. It seems incongruous to apply severe testing in the form of null hypothesis decision criteria and then to be satisfied with merely summarizing the proportions of those very tests. However, the effect size is large and this method does not apparently engender a systemic bias, so it is not problematic in this context.

RESPONSE: We're using this approach for this project because of the specific challenges presented to us by scallops. In the future, we hope to see (and perhaps contribute to) the development of statistical approaches better-suited for analyzing the semi-autonomous actions of appendages in animals with distributed motor systems (e.g. echinoderms, cephalopods, and scallops).